# Genetic and Clinical Studies of Peripheral Neuropathies with Three Small Heat Shock Protein Gene Variants in Korea

**DOI:** 10.3390/genes13030462

**Published:** 2022-03-05

**Authors:** Si On Lim, Na Young Jung, Ah Jin Lee, Hee Ji Choi, Hye Mi Kwon, Wonseok Son, Soo Hyun Nam, Byung-Ok Choi, Ki Wha Chung

**Affiliations:** 1Department of Biological Sciences, Kongju National University, 56 Gongjudaehak-ro, Gongju 32588, Korea; sion100482@naver.com (S.O.L.); jny7765@naver.com (N.Y.J.); jhmom1010@naver.com (A.J.L.); heejee0624@naver.com (H.J.C.); sonwonseok@gmail.com (W.S.); 2Department of Neurology, Samsung Medical Center, Sungkyunkwan University School of Medicine, 81 Irwon-ro, Gangnam-gu, Seoul 06351, Korea; huimei.kwon@samsung.com; 3Stem Cell & Regenerative Medicine Institute, Samsung Medical Center, Seoul 06351, Korea; nanan97@naver.com

**Keywords:** Charcot-Marie-Tooth disease type 2, distal hereditary motor neuropathies, *HSPB1*, *HSPB8*, *HSPB3*, Korean

## Abstract

Small heat shock proteins (sHSPs) are ATP-independent chaperones that help correct the folding of denatured proteins and protect cells from stress. Mutations in *HSPB1*, *HSPB8*, and *HSPB3* are implicated in inherited peripheral neuropathies (IPNs), such as Charcot-Marie-Tooth disease type 2 (CMT2) and distal hereditary motor neuropathies (dHMN). This study, using whole exome sequencing or targeted gene sequencing, identified 9 pathogenic or likely pathogenic variants in these three sHSP genes from 11 Korean IPN families. Most variants were located in the evolutionally well conserved α-crystallin domain, except for p.P182S and p.S187L in *HSPB1*. As an atypical case, a patient with dHMN2 showed two compound heterozygous variants of p.R127Q and p.Y142H in *HSPB1*, suggesting a putative case of recessive inheritance, which requires additional research to confirm. Three *HSPB8* variants were located in the p.K141 residue, which seemed to be a mutational hot spot. There were no significant differences between patient groups, which divided by sHSP genes for clinical symptoms such as onset age, severity, and nerve conduction. Early-onset patients showed a tendency of slightly decreased sensory nerve conduction values compared with late-onset patients. As a first Korean IPN cohort study examining sHSP genes, these results will, we believe, be helpful for molecular diagnosis and care of patients with CMT2 and dHMN.

## 1. Introduction

Inherited peripheral neuropathies (IPNs) are a group of genetically and clinically heterogeneous neuromuscular disorders characterized by distal muscle weakness and loss of sensation in the upper and lower extremities. They include hereditary motor and sensory neuropathy (HMSN), conventionally called Charcot-Marie-Tooth disease (CMT), distal hereditary motor neuropathies (dHMN), hereditary sensory autonomic neuropathies (HSAN), and hereditary neuropathy with pressure palsies (HNPP). As the most common IPN, CMT is usually divided into three categories according to the peripheral nerve conduction velocities (NCVs): the demyelinating type (also called CMT1), the axonal defective type (also called CMT2), and the intermediate type.

Mutations in more than 130 genes with variable functions have been reported to be implicated in the pathogenesis of IPNs. Among these genes, three small heat shock protein (sHSP) genes belonging to the ATP-independent molecular chaperones family have been reported to cause several types of IPNs, such as dominant CMT2 and dHMN. Mutations in *HSPB1* (MIM 602195), which encodes HSP27, are implicated in dominant CMT2F (MIM 606595) and dHMN2B (MIM 608634) [1,2], but rarely associated with the recessive type [2,3]. *HSPB8* (MIM 608014), which encodes HSP22, is implicated in CMT2L (MIM 608673) and dHMN2A (MIM 158590) [4,5]. *HSPB3* (MIM 604624) encoding HSP27-like protein (HSPL27), which is the smallest protein in the sHSP family, is implicated in CMT2 and dHMN2C (MIM 613376) [6,7]. Mutations in *HSPB1* and *HSPB8* were first reported in 2004 as the causes of CMT2 or dHMN [1,4]; more than 40 mutations of three sHSP genes have been reported to cause IPNs [8,9,10]. Mutations in other sHSP members, *HSPB4* and *HSPB5*, are associated with several nervous disorders, although no association with IPN has been reported [11,12].

The sHSPs proteins are involved in a variable cellular process and play an important role in maintaining cellular homeostasis and protecting cells from stress [13,14]. They also interact with matrix proteins to stabilize and prevent misfolding or aggregation of proteins [15]. Dysfunctions of sHSPs have been reported to be involved in various neurodegenerative diseases, such as Alzheimer disease, Parkinson disease, and Huntington disease, caused by protein misfolding, abnormal aggregation, and accumulation [16,17].

We performed this study to identify pathogenic mutations in sHSP genes from Korean patients with IPN. We have previously reported three sHSP mutations in five Korean IPN families [7,18,19,20]. Including the previous results, this study separated nine pathogenic or likely-pathogenic sHSP variants in 11 IPN families. For the patients with sHSP variants, we characterized clinical phenotypes and analyzed genotype–phenotype correlations.

## 2. Materials and Methods

### 2.1. Subjects

This study was conducted using 782 Korean IPN families. Of them, 104 families were diagnosed to be CMT type 1A with *PMP22* duplication, and the remaining 678 families, which consisted of 1016 affected and 944 unaffected individuals, were further examined to determine genetic causes. Paternity was determined for families with de novo mutations by using the PowerPlex Fusion System (Promega, Madison, WI, USA). All participants involved in this study provided written informed consent. For the minors under the ages of 18 years old, consent was provided by their parents. This study was approved by the Institutional Review Boards of Kongju National University (KNU_IRB_2018-27) and Samsung Medical Center (2018-05-102-002).

### 2.2. Clinical Examinations

Clinical symptoms of motor and sensory impairments, muscle atrophies, and deep tendon reflexes (DTRs) were measured as per standard methods. Strengths of the flexor and extensor muscles were manually evaluated by the standard Medical Research Council (MRC) scale. Physical disability was determined by dual methods of the CMT neuropathy score ver. 2 (CMTNSv2) and functional disability scale (FDS). Sensory impairments were measured by severity and level of pain, vibration, temperature, and position. Onset ages were estimated by asking patient’s ages when a physical symptom first appeared [21].

### 2.3. Nerve Conduction Studies

Motor and sensory conduction velocities were measured by surface stimulation. Motor nerve conduction velocities (MNCVs), and compound muscle action potentials (CMAPs) of the median, ulnar, and radial nerves were measured by stimulation of the elbow and wrist. The peroneal and tibial MNCVs and CMAPs were measured by stimulation of the knee and ankle. Amplitude of CMAP was determined from baseline to negative peak value. Sensory nerve conduction velocities (SNCVs) of the median, ulnar, radial, and sural nerves were determined using the finger–wrist segments by the orthodromic method. Amplitude of sensory nerve action potential (SNAP) was determined from positive to negative peaks.

### 2.4. DNA Extraction and Genetic Screening

Total DNA was extracted from blood using the HiGene Genomic DNA Prep Kit (Biofact, Daejeon, Korea). All samples were first examined for *PMP22* duplication, and then, proband samples with no *PMP22* duplication were screened by whole exome sequencing (WES; 512 samples) or targeted gene panel sequencing (TS; 166 samples) based on the next generation sequencing (NGS). Exome capture and NGS were carried out using the SureSelect Human All Exon 50M kit (Agilent Technologies, Santa Clara, CA, USA) and the HiSeq 2000 or 2500 Genome Analyzers (Illumina, San Diego, CA, USA), respectively. TS was performed by the method of Nam et al. [22]. As the reference sequence, the UCSC assembly GRCh37/hg19 was used for mapping (http://genome.ucsc.edu/ accessed on 1 November 2021). Rare or unreported variants with the minor allele frequencies (MAFs) of <0.01 were isolated from IPN-related genes; their MAFs were obtained from human genome databases of the 1000 Genomes Project (1000G; http://www.1000genomes.org/ accessed on 1 November 2021) and Genome Aggregation Database (gnomAD; https://gnomad.broadinstitute.org/ accessed on 1 November 2021). For Korean frequency data, Korean Reference Genome Database (KRGDB; http://coda.nih.go.kr/coda/KRGDB/index.jsp/ accessed on 1 November 2021) was used. Pathogenicity of variants was evaluated basically by the American College of Medical Genetics and Genomics (ACMG) guideline (https://wintervar.wglab.org/ accessed on 1 November 2021) and previous studies. For pathogenic candidate variants, Sanger sequencing was performed to confirm their presence using the SeqStudio or 3130XL Genetic Analyzers (Life Technologies-Thermo Fisher Scientific, Carlsbad, CA, USA).

### 2.5. Conservation Analysis and in Silico Prediction

Amino acid sequence conservation of the mutation sites was determined using MEGA-X program (ver. 5.05; http://www.megasoftware.net/ accessed on 1 November 2021). Mutation effects were predicted using three in silico analysis programs: PolyPhen-2 (http://genetics.bwh.harvard.edu/pph2/ accessed on 1 December 2021), PROVEAN (http://provean.jcvi.org/ accessed on 1 December 2021), and MUpro (http://mupro.proteomics.ics.uci.edu/ accessed on 1 December 2021). Protein 3D structures were predicted using I-TASSER (https://zhanglab.ccmb.med.umich.edu/I-TASSER/ accessed on 1 December 2021), and were visualized using the Mol* feature of the Protein Data Bank (http://www.rcsb.org/ accessed on 1 December 2021). The secondary structure of genomic DNA was predicted using the mFold algorithm (http://www.unafold.org/ accessed on 1 December 2021).

### 2.6. Statistical Analysis

Non-parametric Mann–Whitney U test and parametric unpaired t-test were used to compare the clinical data of patients grouped by sHSP genes. Parametric Pearson r or non-parametric Spearman r values were used to analyze the correlation between clinical values and onset ages. Significant differences were determined at *p* < 0.05. Statistical analyses were performed using GraphPad Prism (ver. 8.00, GraphPad Software, San Diego, CA, USA).

## 3. Results

### 3.1. Causative Variants in sHSP Genes

We identified nine pathogenic or likely pathogenic sHSP gene variants in 11 families from the Korean IPN cohort study (Table 1); their pathogenicity was basically determined using the guideline of ACMG (Appendix A). Five *HSPB1* variants were observed in six families (Figure 1A) and three *HSPB8* variants were observed in four families (Figure 1B). Only one variant was found in *HSPB3* (Figure 1C). These results include the sHSP gene variants in Koreans previously reported by our research group [7,18,19,20]. The genotypes are provided at the bottom of all the examined individuals in the pedigrees in Figure 1. All the causative variants were confirmed in the extended family members by Sanger sequencing (Figure 2A). Amino acids of most variant sites were highly conserved from worm to mammal (Figure 2B). Moreover, variants located in the α-crystallin domain were also fairly conserved among human sHSP paralogues, while the variants located in the variable C-terminal domain were less conserved (Figure 2C). Most variant sites were located in the evolutionally well conserved α-crystallin domains, while the p.P182S and p.S187L in *HSPB1* were located in the C-terminal domain (Figure 3A). Pathogenic effects were predicted for all the causative variants by at least one of three in silico analyses (Table 1).

#### 3.1.1. Variants in *HSPB1*

Five pathogenic or likely pathogenic mutations were identified in *HSPB1* from six CMT2 or dHMN families (Figure 1A). A c.404C>T (p.S135F) was observed three times in one large dHMN family (FC189) and two CMT2 families (FC522 and FC567); many studies have reported this mutation as a genetic cause of CMT2F and dHMN2B [1,2,18,19,23,24]. A 6-year-old boy (VI-2) in the FC189 family and two girls (6 and 7 years old, III-1 and III-2) in the FC522 family had the p.S135F variant, but they were still not affected. As a same site mutation, c.404C>A (p.S135Y) was once reported as an underlying cause of CMT2 [28]. A variant of c.544C>T (p.P182S) was observed in a CMT2F family (FC313). Neither of the unaffected parents (I-1 and I-2) of the proband showed that variant, suggesting a de novo mutation. This variant was once reported by Kijima et al. [25]. A c.560C>T (p.S187L) was observed in a CMT2F family (FC1150), which was reported by Echaniz-Laguna et al. [23]. Two heterozygous variants of c.380G>A (p.R127Q) and c.424T>C (p.Y142H) were interestingly found in a sporadic patient with dHMN (FC1005). The p.R127Q was reported as a variant of uncertain significance in ClinVar (https://www.ncbi.nlm.nih.gov/clinvar/ accessed on 5 December 2021), while two mutations at the p.R127 residue have been reported several times to be pathogenic in Belgian, French, Norwegian, and Japanese CMT2F or dHMN2B patients with a dominant manner: p.R127W [1,23,26,29,30] and p.R127L [31,32]. The p.Y142H was an unreported novel variant. Both variants were evaluated as “likely pathogenic” by the guidelines of the ACMG. Since three children had only one of the two variants (III-1: p.Y142H; III-2 and -3: p.R127Q), each variant seemed to be inherited from each of the proband’s parents. Both parents were normal by history taking and three children showed no abnormal phenotype at the recent neurological examination, including motor and sensory function, and DTR. Thus, it was suggested that the dHMN phenotype might be due to the biallelic effect of the two compound heterozygous variants.

#### 3.1.2. Variants in *HSPB8*

Three variants, c.421A>G (p.K141E), c.422A>C (p.K141T), and c.423G>T (p.K141N), were identified in *HSPB8* from the CMT2 and dHMN families (Figure 1B). Variants of p.K141E and p.K141N, which were identified in three dHMN families (FC585 and FC1196 with p.K141E, and FC107 with p.K141N), have been reported many times as the underlying cause of CMT2L and dHMN2A [4,5,23,26,27]. The p.K141T shown in the CMT2 family (FC031) has been reported previously to be likely pathogenic [20]. In the FC031 family, neither of the unaffected parents showed that variant, suggesting a de novo mutation. So far, four mutations at the p.K141 residue in HSPB8 have been reported as the genetic causes of CMT2 or dHMN: c.421A>G (p.K141E), c.422A>T (p.K141M), c.423G>C (p.K141N), and c.423G>T (p.K141N) [4,5,23,26,27]. Thus, the p.K141 site seems to be a mutational hot spot, which may be related to its location at the stem of a predicted single strand DNA hairpin structure (Figure 3B).

#### 3.1.3. Variant in *HSPB3*

One variant of c.352T>C (p.Y118H) was observed from *HSPB3* in a CMT2 family (FC702) (Figure 1C). The ACMG guideline evaluated the p.Y118H as a variant of uncertain significance (VUS). However, this study classified it as a likely pathogenic variant, because it was previously reported as an underlying cause of CMT2 by Nam et al (7). In *HSPB3*, only a p.R7S mutation was reported as a pathogenic variant causing dHMN2C, except for p.Y118H [6].

### 3.2. Simulation of 3D Protein Structure

Three-dimensional conformational changes of mutant proteins were simulated to predict the effects of the sHSP gene mutations (Figure 4). In *HSPB1*, the p.R127 residue of the p.R127Q mutation in the α-crystallin domain forms salt bridges with negatively charged p.D107 and p.E125 located in the β4 and β5 strands. However, the salt bridge with p.D107 was destroyed in the p.R127Q mutant, predicting a conformational modification. The p.S135 forms hydrogen bonds with p.H124 and p.Y133 within the β-strands arranged in parallel. However, both hydrogen bonds were disappeared in the p.S135F mutant. The HSPB1 p.Y142, which is located at the β6–β7 strand in an α-crystallin domain, has a hydrogen bond with a near-located positively charged p.R140 residue. However, it was predicted to break and the β-strand structure was distorted by the p.Y142H mutation. The p.P182S mutation located in the C-terminal IxI/V motif newly forms two hydrogen bonds with an adjacent hydrophilic p.Q175 residue, which might result in a conformational change of the protein quaternary structure. In the p.S187L mutation of the C-terminal domain, hydrogen bonds of p.S187 with p.F185 and p.Q190 were destroyed and new bonds with p.T184 and p.A189 were formed. The p.S187L mutation was previously reported to cause protein aggregation [23]. In *HSPB8*, p.K141 residue in the α-crystallin domain forms hydrogen bonds with p.N101 and p.V121, and forms a cation–π interaction with p.F139. Amino acid substitution with a negatively charged residue (p.K141E) caused the breakage of these hydrogen bonds and formed new hydrogen bonds with N-terminal residues. In the p.K141N mutation, a new hydrogen bond was similarly formed between p.K141N and N-terminal p.R86. In the p.K141T mutant, a hydrogen bond to p.N101 was broken, which was predicted to cause a structural alteration of the β-sheets.

In *HSPB3*, p.Y118 of wild type forms two hydrogen bonds with p.L94, and p.W93 formed a cation–π interaction with p.K119. However, in the p.Y118H mutant, the cation–π interaction was destroyed, predicting an unstable structural opening.

### 3.3. Rare Variants in sHSP Genes with Uncertain Significance

We observed three rare sHSP variants with MAFs less than 0.01 in the 1000G and KRGDB in the sporadic IPN patients who showed phenotypes of CMT1 or HNPP instead of CMT2 or dHMN: p.R27P and p.Q128X in *HSPB1* and p.F79C in *HSPB8*. The p.R27P in *HSPB1* and p.F79C in *HSPB8* were found in each patient with the HNPP phenotype (Appendix A). The p.Q128X stop-gain variant in *HSPB1* was found in a patient with CMT1. These three variant were evaluated by VUS by the ACMG guideline. Several *HSPB1* stop-gain or frameshift mutations, such as p.W16X, p.W45X, p.L58Cfs*52, and p.L191Qfs*36, have been reported to VUS by ClinVar, but some variants (such as p.L58Afs*105, p.A61Rfs*100, and p.Q175X) have been reported to be pathogenic [23,32,33].

### 3.4. Clinical Characterization of Patients with sHSP Gene Mutations

The clinical phenotypes of 29 patients with sHSP gene mutations are shown in Appendix A. The clinical phenotypes of the patients with three gene mutations were largely similar. Weakness and atrophy of muscles began and predominated in the distal portions of legs, and were less severe in the distal upper limbs. Disability in the distal lower limbs varied from mild weakness to complete paralysis. The mean onset age of the patients with the *HSPB1* mutations was 23.6 ± 8.2 years, with a wide range of 11 to 45 years; that of the patients with the *HSPB8* mutations was 18.6 ± 3.9 years, with a relatively narrow range of 13 to 26 years. In the *HSPB1* group, the mean onset ages of the dHMN2B and CMT2F patients were similarly 23.8 ± 7.1 years and 23.1 ± 10.1 years, respectively. In the *HSPB8* group, the mean onset age of the dHMN2A patients was 19.5 ± 3.3 years, and that of the CMT2L patient was 13 years. The mean FDS and CMTNSv2 were 2.7 ± 1.8 and 13.4 ± 8.2 in the *HSPB1* group, and 3.3 ± 1.5 and 15.7 ± 6.0 in the *HSPB8* group, respectively.

Sensory loss was present in 7 of 20 patients with the *HSPB1* mutations, in 1 of 7 patients with the *HSPB8* mutations, and in all patients with the *HSPB3* mutations. Pes cavus was present in 17 of 20 patients of *HSPB1* and in all patients of *HSPB8* and *HSPB3*. As additional symptoms, one CMT2L patient with p.K141T in *HSPB8* (FC031: II-1) showed scoliosis, and a 62-year-old dHMN2B woman with p.S135F in *HSPB1* (FC189: IV-13) showed the most severe physical disability of being wheelchair bound (FDS: 7, CMTNSv2: 31), while a dHMN2B man with compound heterozygous *HSPB1* mutations of p.R127Q and p.Y142H (FC1005: II-5) revealed late onset (45 years) and mild clinical symptoms (FDS: 1, CMTNSv2: 4) compared with other affected individuals.

When clinical phenotypes were compared between *HSPB1* and *HSPB8* patients, patients with *HSPB8* mutations showed slightly earlier onset and more severe physical disability than those with *HSPB1* mutations, but no significant differences were observed in the onset age (*U* = 39.5, *p* = 0.096), FDS (*U* = 51, *p* = 0.293), and CMTNSv2 (t = 0.695, *p* = 0.493) (Figure 5A–C). In addition, there were no significant differences in onset and physical severity between male and female patients (onset: *U* = 97.5, *p* = 0.859; FDS: *U* = 93, *p* = 0.699; CMTNSv2: t = 0.655, *p* = 0.518). When the correlations between onset ages and physical disability levels were examined for all the examined patients, no significant correlations were observed in the values of FDS and CMTNSv2 (FDS: *p* = 0.418, CMTNSv2: *p* = 0.571) (Figure 6A,B).

### 3.5. Electrophysiological Findings of Patients with sHSP Gene Mutations

The electrophysiological values of 28 patients with sHSP gene mutations are shown in Appendix A. The nerve conduction findings of the patients with three gene mutations were largely similar, and there was no significant difference among the three gene groups. All patients showed predominantly decreased motor CMAP amplitudes in lower limbs compared to upper limbs. The mean median CMAP in upper limbs was 11.1 ± 7.4 mV and the mean peroneal CMAP in lower limbs was 0.8 ± 1.7 mV. In the follow-up studies of some patients, the median MNCV was found to decrease at an average of 1.56 ± 2.20 m/s per year, but the decrease rate of the median SNCV was much lower, with an average of 0.04 ± 1.63 m/s per year.

When phenotypes were compared between *HSPB1* and *HSPB8* patients, no significant differences were observed in the median MNCV (*U* = 47, *p* = 0.213) or median SNCV (*U* = 55, *p* = 0.422) (Figure 5D,E). In addition, there were no significant differences in nerve conduction values between male and female patients (median MNCV: *U* = 77.5, *p* = 0.403; median SNCV: *U* = 89, *p* = 0.763). When the correlations between onset age and electrophysiological features were analyzed, early-onset patients showed a tendency of slightly decreased sensory nerve conduction values compared with late-onset patients (median SNCV: *p* = 0.024 and median SNAP: *p* = 0.031); however, other comparisons of motor nerve conduction values showed no significant correlations (median MNCV: *p* = 0.319, and median CMAP: *p* = 0.308) (Figure 6C,D).

## 4. Discussion

Mutations in the *HSPB1*, *HSPB8*, and *HSPB3* genes have been reported to cause autosomal dominant IPNs of CMT2 and dHMN. This study identified nine pathogenic or likely pathogenic sHSP gene mutations from 11 families in the Korean IPN cohort study. As expected, *HSPB1* mutations were most frequently observed in six families with five mutations. *HSPB8* mutations were observed in four families with three mutations, and *HSPB3* mutation was observed only in a family. Three mutations (p.R127Q in *HSPB1*, p.K141T in *HSPB8*, and p.Y118H in *HSPB3*) were unreported in other ethnic groups, although they were registered in dbSNP. Two paternal-originated de novo mutations were observed in each *HSPB1* and *HSPB8*. The de novo mutation was suggested to have originated from the father by haplotyping analysis using adjacent SNPs. The frequencies of IPN families with sHSP gene mutations was determined to be 1.41% in the total unrelated IPN patients and 1.62% in the patients excluding CMT1A (Table 2). When the frequency in the patients excluding CMT1A was compared with other study groups, it was similar with 1.59% of Japanese [10]. It was higher than the frequencies of Han Chinese (0.91%), British (0.83%), American (0.00%), and Brazilian (0.00%), while lower than those of Italian (4.80%) and Spanish (3.94%) [8,9,34,35,36].

As an unusual case, a male patient with dHMN2 interestingly showed two heterozygous mutations of p.R127Q and p.Y142H in *HSPB1*. The p.R127Q has been registered as a variant of uncertain significance in ClinVar, while p.R127W and p.R127L in the p.R127 residue have been reported to be pathogenic for CMT2F or dHMN2B [1,23,26,29,30,31,32]. Both of this patient’s parents were unaffected, although genetic and neurological testing was not performed, and three children having alternatively one of each variant were also unaffected. Thus, the patient seems to be a recessive type of dHMN2B by bialleles of two compound heterozygotes. However, we could not exclude a possibility that any of two variants may role as a dominant genetic cause. As the reasons, (a) the ages of the three children (29, 28, and 22 years old, respectively) may not have reached the onset age yet, considering the proband’s onset of 45 years old, and (b) it may possible that a variant was inherited by a de novo event. In spite of these two mutations, our patient showed late onset and mild symptoms. So far, a few cases of autosomal recessive CMT2 or dHMN2 patients with *HSPB1* mutations have been reported. As a first report, a p.L99M homozygous mutation was reported in a consanguineous Pakistani patient with dHMN [2]. Then, a p.R140G homozygous mutation was reported in an Indian patient with distal vacuolar myopathy and motor neuropathy [3], and pS315F and p.R316L homozygous mutations were identified in the Republic of Cabo Verde and Iranian families with CMT2, respectively [24].

All three *HSPB8* mutations (c.421A>G;p.K141E, c.422A>C;p.K141T, and c.423G>T; p.K141N) observed in this study were, interestingly, located in the p.K141 residue. Including this result, four mutations at the p.K141 residue have been reported as the genetic causes of CMT2 or dHMN: p.K141E, p.K141T, p.K141M, and p.K141N [4,5,20,23,27]. Thus, the “AAG” sequence coding p.K141 seems to be a mutational hot spot, which is predicted to locate a stem of single-stranded DNA hairpin structures. Most patients with *HSPB8* defects showed mutations in the p.K141 site, with the exception of two families with p.P90L and p.N138T. The p.K141 was well conserved with basic amino acids of lysine or arginine among homologues of paralogues as well as orthologues.

The clinical features of the patients with three gene mutations were largely similar, and there were no significant differences for onset age, severity, nerve conduction, or muscle atrophy between patient groups between *HSPB1* and *HSPB8* mutations. Actually, the clinical features of *HSPB1* and *HSPB8* patients were difficult to distinguish in both clinical examination and nerve conduction studies. Therefore, when sHSP patients are suspected, genetic testing can be said to be the most useful. Physical severities measured by FDS and CMTNSv2, electrophysiological values measured by CMAP, and MNCV showed no significant difference by onset age except for SNCV and SNAP. The mean onset age was earlier in CMT2 patients than in dHMN patients, but with no significant difference.

## 5. Conclusions

This IPN cohort study identified nine sHSP gene mutations in 11 families as the underlying causes of CMT2 or dHMN phenotypes. In particular, this study is the first report of a putative patient with autosomal recessive *HSPB1* mutations in Korea. We carefully analyzed genotype–phenotype correlations and also compared the clinical phenotypes according to genes and onset ages. As a first Korean IPN cohort study analyzing sHSP genes, we believe that this study will be helpful for the molecular diagnosis and care of patients with CMT2 and dHMN.

## Figures and Tables

**Figure 1 genes-13-00462-f001:**
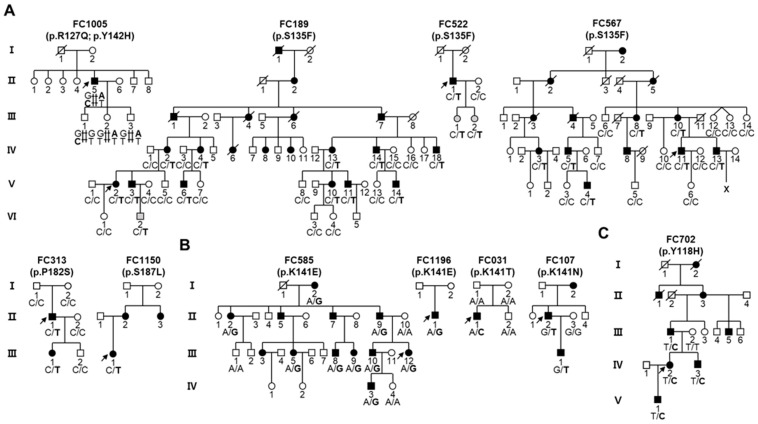
Inherited peripheral neuropathy families with small heat shock protein gene mutations. Genotypes of the interested variants are provided at the bottom of all examined family members. Arrows indicate the proband (unfilled symbols (□, ○): unaffected individuals; black-filled symbols (■, ●): affected individuals; gray-filled symbols: unaffected individuals having corresponding mutation; ˄: twins). (**A**) Pedigrees with *HSPB1* mutations. (**B**) Pedigrees with *HSPB8* mutations. (**C**) Pedigree with *HSPB3* mutation.

**Figure 2 genes-13-00462-f002:**
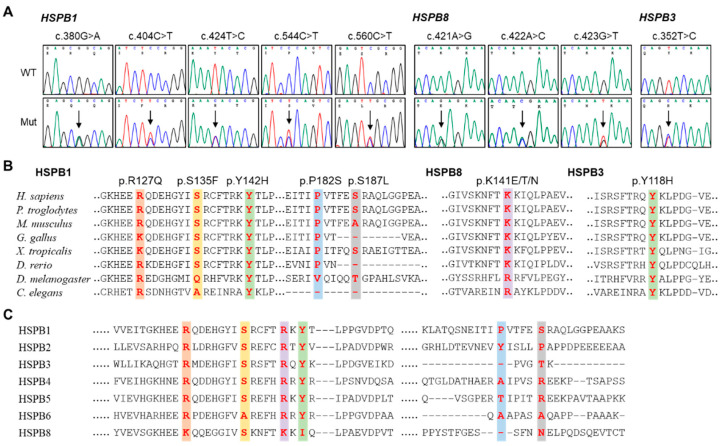
Mutations in three small heat shock protein genes. (**A**) Chromatograms of the mutation sites. They were obtained by Sanger sequencing. Vertical arrows indicate the mutation sites (WT: wild type allele, Mut: mutant allele). (**B**) Conservation of amino acids in the mutation sites (red) and vicinant sequences. Reference sequences are: NP_001531.1 (*Homo sapiens*), XP_519162.3 (*Pan troglodytes*), NP_038588.2 (*Mus musculus*), NP_990621.1 (*Gallus gallus*), NP_001072817.1 (*Xenopus tropicalis*), NP_001008615.2 (*Danio rerio*), NP_001287001.1 (*Drosophila melanogaster*), NP_498776.1 (*Caenorhabditis elegans*) for HSPB1, NP_055180.1 (*H. sapiens*), XP_509417.1 (*P. troglodytes*), NP_109629.1 (*M. musculus*), XP_004934466.1 (*G. gallus*), NP_001005658.1 (*X. tropicalis*), NP_001094427.2 (*D. rerio*), NP_001027115.1 (*D. melanogaster*), NP_498776.1 (*C. elegans*) for HSPB8, NP_006299.1 (*H. sapiens*), XP_517764.2 (*P. troglodytes*), NP_064344.1 (*M. musculus*), XP_001231558.1 (*G. gallus*), XP_002941074.1 (*X. tropicalis*), NP_001092922.1 (*D. rerio*), NP_523999.1 (*D. melanogaster*), NP_498776.1 (*C. elegans*) for HSPB3. (**C**) Amino acid sequence conservation at the mutation sites (red) among human sHSP paralogues. Reference amino acid sequences are: HSPB2: NP_001532.1, HSPB4: NP_000385.1, HSPB5: NP_001276736.1, and HSPB6: NP_653218.1.

**Figure 3 genes-13-00462-f003:**
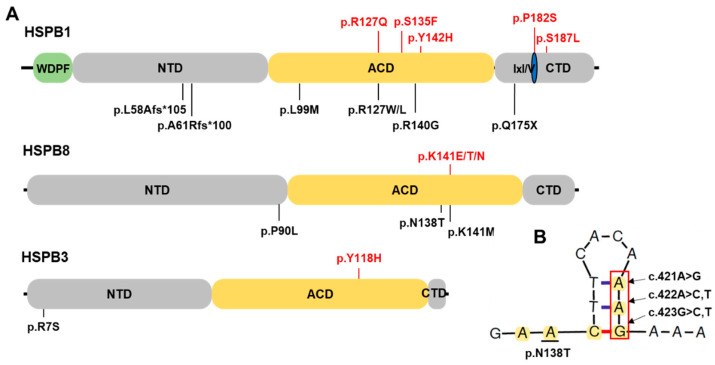
Schemes of three small heat shock proteins and predicted secondary structure of single strand DNA region for p.N138 to p.K141 residues in HSPB8. (**A**) Schematic diagrams of three sHSP proteins. Pathogenic or likely pathogenic variants identified in this study are indicated at the top of the diagrams (red). Some previously reported mutations are indicated at the bottom of the diagrams (ACD: α-crystallin domain, CTD: C-terminal domain, IxI/V: IxI/V motif within CTD, NTD: N-terminal domain, and WDPF: WDPF motif within NTD). (**B**) Predicted secondary structure of single strand DNA region for p.N138 to p.K141 residues in HSPB8. The nucleotides corresponding to p.N138 and p.K141 are indicated in yellow. The p.K141 was distinguished from p.N138 by a red box.

**Figure 4 genes-13-00462-f004:**
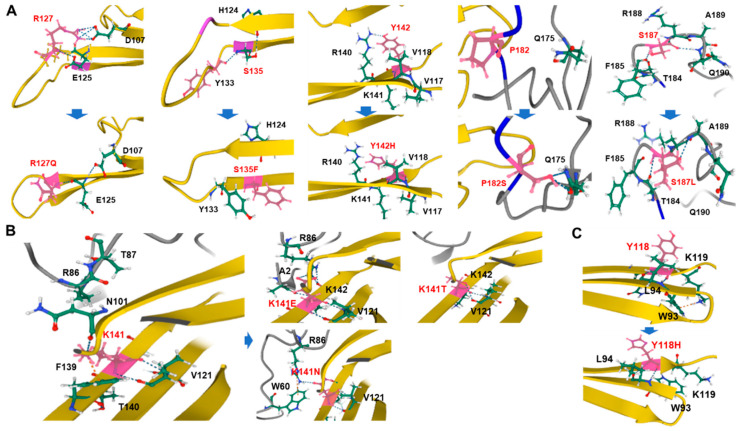
Predicted 3D conformational changes of neighboring structures of causative (pathogenic or likely pathogenic) mutation sites in three small heat shock proteins. Wild type protein structures and their mutant structures are provided up (or left) and down (or right) in each pair. The amino acid sites with mutation are indicated in pink. Hydrogen bonds are indicated by blue dotted lines, and carbon, hydrogen, nitrogen, and oxygen are indicated in green, gray, blue, and red, respectively. Noncovalent molecular interactions (cation–π interaction) are indicated by orange dotted lines. The crystal structures of α-crystallin domains are colored in yellow; N- and C-terminal domains are depicted in gray. Wild and mutant proteins are illustrated by ribbon diagrams. (**A**) HSPB1, (**B**) HSPB8, and (**C**) HSPB3.

**Figure 5 genes-13-00462-f005:**
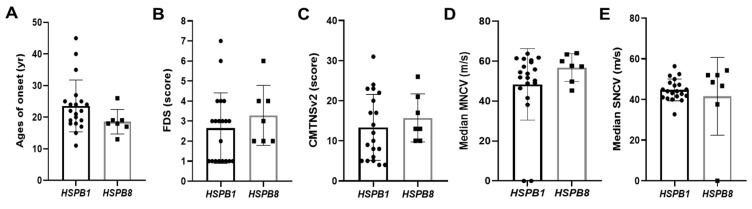
Comparison of clinical and electrophysiological phenotypes between two patient groups having mutations alternatively in *HSPB1* and *HSPB8*. The patients were classified into two groups of *HSPB1* and *HSPB8*, then their phenotypic values were compared. (**A**) Age of onset, (**B**) FDS, (**C**) CMTNSv2, (**D**) median MNCV, and (**E**) median SNCV.

**Figure 6 genes-13-00462-f006:**
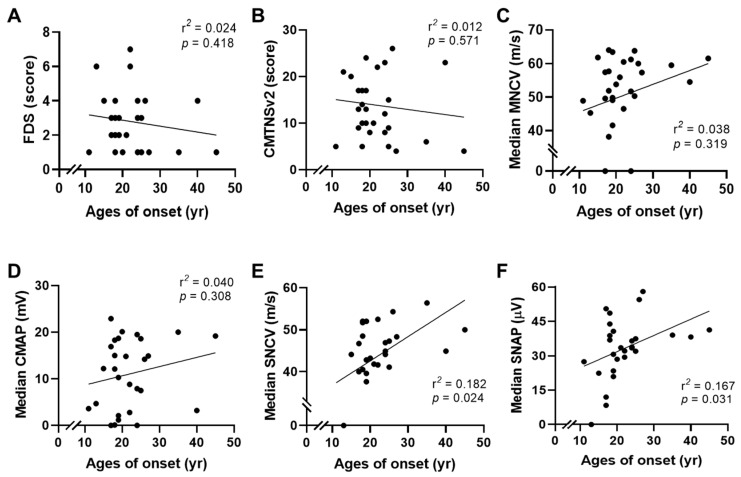
Correlation of clinical phenotypes according to onset ages (OA). (**A**) OA vs. FDS, (**B**) OA vs. CMTNSv2, (**C**) OA vs. median MNCV, (**D**) OA vs. median CMAP, (**E**) OA vs. median SNCV, and (**F**) OA vs. median SNAP.

**Table 1 genes-13-00462-t001:** Disease-causing variants identified from *HSPB1*, *HSPB8*, and *HSPB3* in Korean inherited peripheral neuropathy patients.

Genes	Families		Mutations		Mutant Allele Frequencies ^2^			In Silico Analyses ^3^			Class	Notes and References
	ID	Type	Nucleotide ^1^	Amino acid	1000G	gnomAD	KRGDB	PRO	PP2	MUp		
*HSPB1*	FC1005	dHMN2B	[c.380G>A] +	[p.R127Q] +	NR	4.0×10^−6^	NR	−3.35 *	1.00 *	−0.58 *	LP	Biallelic
			[c.424T>C]	[p.Y142H]	NR	NR	NR	−4.75 *	1.00*	−0.32 *	LP	
	FC189 FC522 FC567	dHMN2B CMT2F CMT2F	c.404C>T	p.S135F	NR	NR	NR	−5.01 *	0.96 *	−0.37 *	P	[1,2,18,19,23,24]
	FC313	CMT2F	c.544C>T	p.P182S	NR	NR	NR	−6.84 *	1.00 *	−1.00 *	P	De novo, [25]
	FC1150	CMT2F	c.560C>T	p.S187L	NR	4.1×10^−6^	NR	−2.17	0.62 *	1.00	P	[23]
*HSPB8*	FC585 FC1196	dHMN2A	c.421A>G	p.K141E	NR	NR	NR	−2.39	1.00 *	0.79	P	[4,26,27]
	FC031	CMT2L	c.422A>C	p.K141T	NR	NR	NR	−3.62 *	1.00 *	0.16	P	De novo, [20]
	FC107	dHMN2A	c.423G>T	p.K141N	NR	NR	NR	−2.63 *	1.00 *	0.22	P	[5,23,26]
*HSPB3*	FC702	CMT2	c.352T>C	p.Y118H	NR	4.0×10^−6^	NR	−4.97 *	1.00 *	−0.85 *	LP	[7]

^1^ Reference DNA sequences: *HSPB1*: NM_001540.5, *HSPB8*: NM_014365.3, and *HSPB3*: NM_006308.3. ^2^ Minor allele frequencies from the 1000 Genomes Project (1000G), the Genome Aggregation Database (gnomAD), and Korean Reference Genome Database (KRGDB). ^3^ In silico scores of PolyPhen-2 (PP2) ~1, PROVEAN (PRO) < −2.5, and MUpro (MU) < 0 indicate pathogenic prediction (* denotes a pathogenic prediction). Abbreviations: CMT, Charcot-Marie-Tooth disease; dHMN, distal hereditary motor neuropathy; LP, likely pathogenic; NR, nonreported; P, pathogenic.

**Table 2 genes-13-00462-t002:** Frequencies of inherited peripheral neuropathy patients with small heat shock protein gene mutations according to populations.

Populations	Sample Numbers		sHSP genes			Frequencies (%)		References
	Total	CMT1A Exclusion	*HSPB1*	*HSPB8*	*HSPB3*	Total	CMT1A Exclusion	
Korean	782	678	6	4	1	1.41	1.62	This study
Han Chinese (Taiwan)	427	219	2	0	0	0.47	0.91	[9]
Italian	566	333	14	2	0	2.83	4.80	[34]
Japanese	-	1005	14	1	1	-	1.59	[10]
British (London)	-	120	0	1	0	-	0.83	[8]
American (Iowa)	-	100	0	0	0	-	0.00	[8]
Brazilian	503	387	0	0	0	0.00	0.00	[36]
Spanish	438	254	7	3	0	2.28	3.94	[35]

Abbreviations: sHSP, small heat shock protein; CMT1A, Charcot-Marie-Tooth disease type 1A.

## Data Availability

The data presented in this study are available in article. Additional WES raw data are available upon request to corresponding authors.

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
