# Peer review of "Genetic and Clinical Studies of Peripheral Neuropathies with Three Small Heat Shock Protein Gene Variants in Korea"

_genes, 2022, doi:10.3390/genes13030462_

Round 1

Reviewer 1 Report

In the manuscript entitled “Genetic and Clinical Studies of Peripheral Neuropathies with 2 Three Small Heat Shock Protein Gene Variants in Korea”, Lim et al. reported 9 pathogenic or likely pathogenic variants in HSPB1, HSPB8, or HSPB3 genes from 11 Korean families with inherited neuropathy. They also presented comprehensive and delicate analysis of the genotype, clinical phenotype and electrophysiological findings of the patients. Because the pathogenic mutations of HSPB1, HSPB8, or HSPB3 are rare, it is difficult to recruit such a large cohort. The manuscript was well written and the information provided by this manuscript is interesting and valuable. I have no any major suggestion. Only a typo, the second word in line 305, “that” should be  revised to  “than”.

Author Response

We sincerely thank you for your favorable comments on our manuscript. As your comment, “that” was revised to “than”. In addition to your comments, we carefully revised the manuscript.

Reviewer 2 Report

The authors present variants in three sHSP genes from 11 Korean IPN families. Of course to further knowledge about these diseases, articles as the one presented by the authors are valuable. However, taking into account the previous 4 publications of the authors, in which 4 families from this article have already been described in detail, I would like to see more novelty in this study.

So, I have a few concerns that should be addressed:

1)           Line 104: You write «Some samples were applied for targeted gene sequencing including three sHSP genes». If I understand correctly, some samples were tested by using Sanger sequencing. Please write about the criteria for selecting these patients and the number of patients tested.

2)           Line 104: How many patients were tested by using exome sequencing? Please, write about it.

3)           A small number of patients with mutations in the HSPB3 gene have been described in the world. Please describe if there are any new clinical features in patients from the FC 702 family compared to previous cases?

4)           I have a few questions about interpretation of the variants according to the ACMG:

-             Why do you use the PP4 criterion? Phenotype of inherited peripheral neuropathies is not highly specific. IPNS is group of genetically heterogeneous disorders. Mutations in more than 100 genes have been reported to be implicated in the pathogenesis of IPNs.

-             You use PS4 criterion. Please, describe, how did you calculate OR for c.544C>T, c.560C>T, c.422A>C and c.352T>C.

-             Why do you use the PM1 criterion for variant c.352T>C in HSPB3? Why did you decide that this variant is a mutational hotspot, or is localized in a critical and well-established functional domain? In addition, doubt is caused by the fact that benign variants are described in nearby codons (for example, p.Arg116Gln p.Lys119Glu).

5)           About FC1005 family. You suggested that the proband’s phenotype is due to the biallelic effect of the two compound heterozygous variants since the three unaffected children of the proband were found to have only one of the two variants. Variable age at onset dominant neuropathy caused by variants in the HSPB1 gene is range 15 to 60 years. How old are proband's children? Could it be that one of the variants is autosomal dominant, and the other has no clinical significance? Two mutations at the p.R127 residue have been reported many times to be pathogenic. Under what type of inheritance were they described?  Please discuss this in the article.

6)           Of particular interest are always previously undescribed variants that are the cause of the disease. Tell me, do you have the opportunity to clarify the pathogenicity of the uncertain significant variants (c.382C>T and c.236T>G)? It would add to the value of this manuscript.

Overall, I think that the authors have done a big and very important job. But I feel it is important to publish new mutations with clinical findings and would therefore be happy to see additional evidence of pathogenicity of previously undescribed variants.­­­­­­­­

Author Response

The authors present variants in three sHSP genes from 11 Korean IPN families. Of course to further knowledge about these diseases, articles as the one presented by the authors are valuable. However, taking into account the previous 4 publications of the authors, in which 4 families from this article have already been described in detail, I would like to see more novelty in this study.

families from this article have already been described in detail, I would like to see more novelty in this study.

So, I have a few concerns that should be addressed:

1) Line 104: You write «Some samples were applied for targeted gene sequencing including three sHSP genes». If I understand correctly, some samples were tested by using Sanger sequencing. Please write about the criteria for selecting these patients and the number of patients tested.

[Answer]

Thank you very much for your comments. Targeted gene sequencing was performed by a NGS-based method using a gene panel including 73 IPN related genes, which was reported by Nam et al (2016). Targeted gene sequencing was applied to 166 samples, which were selected with no specific criteria. As your comments, additional explanation and reference (Nam et al. 2016) was provided for the targeted sequencing as followed: Then, proband samples with no PMP22 duplication were tested by whole exome sequencing (WES; 512 samples) or targeted gene panel sequencing (TS; 166 samples) based on the next generation sequencing (NGS). …... TS was performed by the method of Nam et al [22].

2) Line 104: How many patients were tested by using exome sequencing? Please, write about it

[Answer]

Thank you very much for your comment. 512 proband samples were tested by WES, and it was provided in the text: …. by whole exome sequencing (WES; 512 samples) ….

3) A small number of patients with mutations in the HSPB3 gene have been described in the world. Please describe if there are any new clinical features in patients from the FC 702 family compared to previous cases?

[Answer]

Thank you very much for your comment pointing out very important issue. As the cases of HSPB1 and HSPB8, mutations inHSPB3 cause CMT2 or dHMN. The clinical features of the patients with three gene mutations were largely similar, and there were no significant differences for onset age, severity, nerve conduction, or muscle atrophy among patient groups divided by genes. Since only a single family including 2 patients had a HSPB3 mutation, statistical significance for phenotypes by gene groups was performed only between the HSAPB1 and HSPB8 groups, excluding the HSPB3 patients (Figure 5). Details of clinical phenotypes and electrophysiological values are provided in Table S3 and S3.

4) I have a few questions about interpretation of the variants according to the ACMG:

- Why do you use the PP4 criterion? Phenotype of inherited peripheral neuropathies is not highly specific. IPNS is group of genetically heterogeneous disorders. Mutations in more than 100 genes have been reported to be implicated in the pathogenesis of IPNs.

[Answer]

Thank you very much for your comments. As you mentioned, IPNs are known as a group of clinically and genetically heterogeneous disorders with a loose genotype-phenotype correlation. However, each IPN family exhibits Mendelian inheritance with a single genetic etiology. In addition, all the affected individuals with sHSP mutations showed CMT2 or dHMN types. Thus, we considered that CMT2 or dHMN patients with sHSP mutations met the PP4 criterion.

- You use PS4 criterion. Please, describe, how did you calculate OR for c.544C>T, c.560C>T, c.422A>C and c.352T>C.

[Answer]

Thank you very much for your comments. This study was not corresponded to case–control study, thus OR or statistically significance could not be calculated. Thus, we excluded the PS4 criterion and reevaluated corresponding variants in Table S1.

- Why do you use the PM1 criterion for variant c.352T>C in HSPB3? Why did you decide that this variant is a mutational hotspot, or is localized in a critical and well-established functional domain? In addition, doubt is caused by the fact that benign variants are described in nearby codons (for example, p.Arg116Gln p.Lys119Glu).

[Answer]

Thank you very much for your comments. The p.Y118 in HSPB3 did not locate in a mutational hotspot. But the PM1 criterion was applied, because it locates in the highly conserved α-crystallin domain among sHSPs. Also, the p.Y118H in HSPB3 corresponds to the homologous site of the p.Y142H in HSPB1.

5) About FC1005 family. You suggested that the proband’s phenotype is due to the biallelic effect of the two compound heterozygous variants since the three unaffected children of the proband were found to have only one of the two variants. Variable age at onset dominant neuropathy caused by variants in the HSPB1 gene is range 15 to 60 years. How old are proband's children? Could it be that one of the variants is autosomal dominant, and the other has no clinical significance? Two mutations at the p.R127 residue have been reported many times to be pathogenic. Under what type of inheritance were they described? Please discuss this in the article.

[Answer]

Thank you very much for your comments. Proband's onset was 45 years old, and ages of three children are 29, 28, and 22 years old, respectively. Like your concern, they may not reach onset age yet. When neurological examination including motor and sensory function, and deep tendon reflex (DTR) was performed in November last year, all three children were normal in all items. The muscle strength of the lower and upper extremities was normal with an MRC score of 5, and there was no case of sensory loss in the pin prik and vibration sense. In the DTR test, the knee jerk, ankle jerk, biceps jerk, and triceps jerk were all normal with the grade of ++. In many IPN cases with late onset, DTR was often abnormal even before clinical symptoms appeared, but all three children were normal. In particular, both parents of the proband were neurologically normal by history taking, although no genetic test or neurological examination were performed. Therefore, we assumed that this patient was a recessive case caused by biallelic variants. However, it may possible that one of two mutations was transmitted by a de novo mutation. Thus, as your comments, the sentences of “Since three children had only one of the two variants (III-1: p.Y142H; III-2 and -3: p.R127Q), each variant was seemed to be transmitted from the proband's each parent. Both parents were normal by history taking and three children showed no abnormal phenotype at the recent neurological examination including motor and sensory function, and DTR.” were added in the Result section, and the sentences of “However, we could not exclude a possibility that any of two variants may role as a dominant genetic cause. As the reasons, (a) ages of three children (29, 28, and 22 years old, respectively) may not reach the onset age yet, considering the proband's onset of 45 years old, and (b) it may possible that a variant was inherited by a de novo event.” were added in the Discussion section. 

6) Of particular interest are always previously undescribed variants that are the cause of the disease. Tell me, do you have the opportunity to clarify the pathogenicity of the uncertain significant variants (c.382C>T and c.236T>G)? It would add to the value of this manuscript. Overall, I think that the authors have done a big and very important job. But I feel it is important to publish new mutations with clinical findings and would therefore be happy to see additional evidence of pathogenicity of previously undescribed variants.

[Answer]

Thank you very much for your comments. For the sporadic CMT1 patient (FC371) with HSPB1 c.382C>T (p.Q128X), (a) PP4 criterion was excluded from ACMG because of CMT1 phenotype (finally evaluated as “VUS” by ACMG), and (b) many stop-gain frameshift mutations were evaluated to VUS, such as p.W16X, p.W45X, p.L58Cfs*52, and p.L191Qfs*36. For the sporadic hereditary neuropathy with liability to pressure palsies (HNPP) patient (HN104), PP4 criterion was excluded from ACMG because of HNPP phenotype (finally evaluated as “VUS” by ACMG). Like your concerns, we have been thinking a lot about these two variants. However, as both patients were sporadic cases, with further genetic testing of family members blocked, we had no choice but to judge them as "VUS". We revised or added following sentences: “We observed three sHSP variants of uncertain significannce (VUSs) with MAFs less than 0.01 in the 1000G and KRGDB in the sporadic IPN patients who showed phenotypes of CMT1 or HNPP instead of CMT2 or dHMN.” And “These three variant were evaluated by VUS by the ACMG guideline.”

Round 2

Reviewer 2 Report

Dear authors. Thank you for the corrections. Below I will write those points in which I still have questions.

4) - Why do you use the PP4 criterion? Phenotype of inherited peripheral neuropathies is not highly specific. IPNS is group of genetically heterogeneous disorders. Mutations in more than 100 genes have been reported to be implicated in the pathogenesis of IPNs.

[Answer] Thank you very much for your comments. As you mentioned, IPNs are known as a group of clinically and genetically heterogeneous disorders with a loose genotype-phenotype correlation. However, each IPN family exhibits Mendelian inheritance with a single genetic etiology. In addition, all the affected individuals with sHSP mutations showed CMT2 or dHMN types. Thus, we considered that CMT2 or dHMN patients with sHSP mutations met the PP4 criterion.

[Reviewer’s answer]

You write «all the affected individuals with sHSP mutations showed CMT2 or dHMN types». Please, write down how many genes are responsible for the development of CMT2 or dHMN? Below I wrote a quote from ACMG about pp4 criteria, please describe what criteria your options match.

PP4: In general, the fact that a patient has a phenotype that matches the known spectrum of clinical features for a gene is not considered evidence for pathogenicity given that nearly all patients undergoing disease-targeted tests have the phenotype in question. If the following criteria are met, however, the patient’s phenotype can be considered supporting evidence: (i) the clinical sensitivity of testing is high, with most patients testing positive for a pathogenic variant in that gene; (ii) the patient has a welldefined syndrome with little overlap with other clinical presentations (e.g., Gorlin syndrome including basal cell carcinoma, palmoplantar pits, odontogenic keratocysts); (iii) the gene is not subject to substantial benign variation, which can be determined through large general population cohorts (e.g., Exome Sequencing Project); and (iv) family history is consistent with the mode of inheritance of the disorder. PP5 BP6 reputable source There are increasing examples where pathogenicity classifications from a reputable source (e.g., a clinical laboratory with long-standing expertise in the disease area) have been shared in databases, yet the evidence that formed the basis for classification was not provided and may not be easily obtainable. In this case, the classification, if recently submitted, can be used as a single piece of supporting evidence. However, laboratories are encouraged to share the basis for classification as well as communicate with submitters to enable the underlying evidence to be evaluated and built upon. If the evidence is available, this criterion should not be use

- Why do you use the PM1 criterion for variant c.352T>C in HSPB3? Why did you decide that this variant is a mutational hotspot, or is localized in a critical and well-established functional domain? In addition, doubt is caused by the fact that benign variants are described in nearby codons (for example, p.Arg116Gln p.Lys119Glu).

[Answer] Thank you very much for your comments. The p.Y118 in HSPB3 did not locate in a mutational hotspot. But the PM1 criterion was applied, because it locates in the highly conserved α-crystallin domain among sHSPs. Also, the p.Y118H in HSPB3 corresponds to the homologous site of the p.Y142H in HSPB1.

[Reviewer’s answer]

In ACMG criteria said that «PM1 mutational hot spot and/or critical and wellestablished functional domain Certain protein domains are known to be critical to protein function, and all missense variants in these domains identified to date have been shown to be pathogenic. These domains must also lack benign variants. In addition, mutational hotspots in less well-characterized regions of genes are reported, in which pathogenic variants in one or several nearby residues have been observed with greater frequency. Either evidence can be considered moderate evidence of pathogenicity».

They write that «These domains must also lack benign variants.» Also I said earlier that benign variants are described in nearby codons (for example, p.Arg116Gln p.Lys119Glu).

5) About FC1005 family. You suggested that the proband’s phenotype is due to the biallelic effect of the two compound heterozygous variants since the three unaffected children of the proband were found to have only one of the two variants. Variable age at onset dominant neuropathy caused by variants in the HSPB1 gene is range 15 to 60 years. How old are proband's children? Could it be that one of the variants is autosomal dominant, and the other has no clinical significance? Two mutations at the p.R127 residue have been reported many times to be pathogenic. Under what type of inheritance were they described? Please discuss this in the article.

[Answer] Thank you very much for your comments. Proband's onset was 45 years old, and ages of three children are 29, 28, and 22 years old, respectively. Like your concern, they may not reach onset age yet. When neurological examination including motor and sensory function, and deep tendon reflex (DTR) was performed in November last year, all three children were normal in all items. The muscle strength of the lower and upper extremities was normal with an MRC score of 5, and there was no case of sensory loss in the pin prik and vibration sense. In the DTR test, the knee jerk, ankle jerk, biceps jerk, and triceps jerk were all normal with the grade of ++. In many IPN cases with late onset, DTR was often abnormal even before clinical symptoms appeared, but all three children were normal. In particular, both parents of the proband were neurologically normal by history taking, although no genetic test or neurological examination were performed. Therefore, we assumed that this patient was a recessive case caused by biallelic variants. However, it may possible that one of two mutations was transmitted by a de novo mutation. Thus, as your comments, the sentences of “Since three children had only one of the two variants (III-1: p.Y142H; III-2 and -3: p.R127Q), each variant was seemed to be transmitted from the proband's each parent. Both parents were normal by history taking and three children showed no abnormal phenotype at the recent neurological examination including motor and sensory function, 4 and DTR.” were added in the Result section, and the sentences of “However, we could not exclude a possibility that any of two variants may role as a dominant genetic cause. As the reasons, (a) ages of three children (29, 28, and 22 years old, respectively) may not reach the onset age yet, considering the proband's onset of 45 years old, and (b) it may possible that a variant was inherited by a de novo event.” were added in the Discussion section.

[Reviewer’s answer]

Have you tested this patient with Sanger sequencing or NGS? How did your understand that these variants are compound heterozygous?

Please correct the phrase in abstract "As an atypical case, a patient with dHMN2 showed two compound heterozygous variants of p.R127Q and p.Y142H in HSPB1, suggetting recessive inheritance" so that it is clear that you are not claiming that recessive inheritance is observed in this case and that additional research is needed.

Also you didn’t answer my question « Two mutations at the p.R127 residue have been reported many times to be pathogenic. Under what type of inheritance were they described? Please discuss this in the article.»

6) Of particular interest are always previously undescribed variants that are the cause of the disease. Tell me, do you have the opportunity to clarify the pathogenicity of the uncertain significant variants (c.382C>T and c.236T>G)? It would add to the value of this manuscript. Overall, I think that the authors have done a big and very important job. But I feel it is important to publish new mutations with clinical findings and would therefore be happy to see additional evidence of pathogenicity of previously undescribed variants.

[Answer] Thank you very much for your comments. For the sporadic CMT1 patient (FC371) with HSPB1 c.382C>T (p.Q128X), (a) PP4 criterion was excluded from ACMG because of CMT1 phenotype (finally evaluated as “VUS” by ACMG), and (b) many stop-gain frameshift mutations were evaluated to VUS, such as p.W16X, p.W45X, p.L58Cfs*52, and p.L191Qfs*36. For the sporadic hereditary neuropathy with liability to pressure palsies (HNPP) patient (HN104), PP4 criterion was excluded from ACMG because of HNPP phenotype (finally evaluated as “VUS” by ACMG). Like your concerns, we have been thinking a lot about these two variants. However, as both patients were sporadic cases, with further genetic testing of family members blocked, we had no choice but to judge them as "VUS". We revised or added following sentences: “We observed three sHSP variants of uncertain significannce (VUSs) with MAFs less than 0.01 in the 1000G and KRGDB in the sporadic IPN patients who showed phenotypes of CMT1 or HNPP instead of CMT2 or dHMN.” And “These three variant were evaluated by VUS by the ACMG guideline.”

[Reviewer’s answer]

Very sorry that you cannot define the pathogenicity of variants of uncertain clinical significance. Of course, I understand that sometimes it is very difficult to test parents. But all of the pathogenic and probably pathogenic variants have already been described earlier, including by you in 4 articles (except variants in FC1005, the recessive pathogenicity of which is not defined due to the absence of parental testing). Please, can you clarify what is new in your manuscript article?

Author Response

You write «all the affected individuals with sHSP mutations showed CMT2 or dHMN types». Please, write down how many genes are responsible for the development of CMT2 or dHMN? Below I wrote a quote from ACMG about pp4 criteria, please describe what criteria your options match.

[Answer] Thank you very much for your comments. We determined that our cases met to the supporting evidence of (iv) family history is consistent with the mode of inheritance of the disorder. Thus, we considered that CMT2 or dHMN patients with sHSP mutations met the PP4 criterion.

In ACMG criteria said that «PM1 mutational hot spot and/or critical and well established functional domain Certain protein domains are known to be critical to protein function, and all missense variants in these domains identified to date have been shown to be pathogenic. These domains must also lack benign variants. In addition, mutational hotspots in less well-characterized regions of genes are reported, in which pathogenic variants in one or several nearby residues have been observed with greater frequency. Either evidence can be considered moderate evidence of pathogenicity». They write that «These domains must also lack benign variants.» Also I said earlier that benign variants are described in nearby codons (for example, p.Arg116Gln p.Lys119Glu).

[Answer] Thank you very much for your comments. We previously used PM1 criterion because it locates in the highly conserved α-crystallin domain among human sHSP paralogues, although several benign variants were reported in the neighboring sites. The α-crystallin domains are evolutionally very conserved from plants to animals and have critical roles for molecular chaperones. But, as your comment, we exclude PM1 criterion from the variant c.352T>C in HSPB3. Please see Table S1.

Have you tested this patient with Sanger sequencing or NGS? How did your understand thatthese variants are compound heterozygous?

Please correct the phrase in abstract "As an atypical case, a patient with dHMN2 showed twocompound heterozygous variants of p.R127Q and p.Y142H in HSPB1, suggesting recessive inheritance" so that it is clear that you are not claiming that recessive inheritance is observed in this case and that additional research is needed.

Also you didn’t answer my question « Two mutations at the p.R127 residue have been reported many times to be pathogenic. Under what type of inheritance were they described? Please discuss this in the article.»

[Answer] Thank you very much for your comments. (1) We didn’t perform DNA testing for the proband’s parents, but, trans (compound heterozygous) arrays of two variants could be predicted by alternative possess of variants of three children. Please see Figure 1A. (2) Since both parents who presumed to have each one of the two mutations were normal, we proposed recessive inheritance by biallelic variants. We inserted words “a putative case of” in the corresponding sentence. In addition, we edited a sentence of “Thus, it could be suggested that the dHMN phenotype is due to the biallelic effect of the two compound heterozygous variants.” into “Thus, it was suggested that the dHMN phenotype might be due to the biallelic effect of ….” [Results: Page 6, line 203-204]. (3) Two variants of p.R127W and p.R127L have been reported to be associated with CMT2F or dHMN2B with dominant manner. Thus, we insert “with dominant manner” in the corresponding sentence [Results; Page 6, line 194-197].

Very sorry that you cannot define the pathogenicity of variants of uncertain clinical significance. Of course, I understand that sometimes it is very difficult to test parents. But all of the pathogenic and probably pathogenic variants have already been described earlier, including by you in 4 articles (except variants in FC1005, the recessive pathogenicity of which is not defined due to the absence of parental testing). Please, can you clarify what is new in your manuscript article?

[Answer] Thank you very much for your comments. We have already reported 4 out of 11 IPN pedigrees, but seven are still unreported pedigrees. Although this study contains only two novel mutations, we provided the clinical and electrophysiological phenotypes of all patients in detail, and compared the clinical differences according to sHSP genes. We think that it may be the largest comprehensive study that analyzes the genetic variants and clinical characteristics of sHSP genes. Therefore, we would like to argue that our study has sufficient novelty and academic values.

Round 3

Reviewer 2 Report

Dear authors, thank you very much for your corrections!
I have only a few comments: 

1) You have removed the PM1 criterion for variant c.352T>C in the HSPB3 gene. In this case, according to the ACMG criteria, this variant is VUS. Please correct it. 

2) Please correct this phrase in abstract: "As an atypical case, a patient with dHMN2 showed two compound heterozygous variants of p.R127Q and p.Y142H in HSPB1, suggesting a putative case of recessive inheritance" to "As an atypical case, a patient with dHMN2 showed two compound heterozygous variants of p.R127Q and p.Y142H in HSPB1, suggesting a putative case of recessive inheritance, which requires additional research to confirm."

Author Response

I have only a few comments:

1) You have removed the PM1 criterion for variant c.352T>C in the HSPB3 gene. In this case, according to the ACMG criteria, this variant is VUS. Please correct it.

[Answer] Thank you very much for your comment. As the comment, we corrected “LP” into VUS in the Table S1.

2) Please correct this phrase in abstract: "As an atypical case, a patient with dHMN2 showed two compound heterozygous variants of p.R127Q and p.Y142H in HSPB1, suggesting a putative case of recessive inheritance" to "As an atypical case, a patient with dHMN2 showed two compound heterozygous variants of p.R127Q and p.Y142H in HSPB1, suggesting a putative case of recessive inheritance, which requires additional research to confirm."

[Answer] Thank you very much for your comment. The corresponding sentence was revised according to the comment (see Abstract).